# Semen amyloids participate in spermatozoa selection and clearance

Nadia R Roan[1,2]*[†], Nathallie Sandi-Monroy[3,4][†], Nargis Kohgadai[1,2][†], Shariq M Usmani[5], Katherine G Hamil[6§], Jason Neidleman[1,2], Mauricio Montano[2], Ludger Ständker[3,7], Annika Röcker[3], Marielle Cavrois[2,8], Jared Rosen[9], Kara Marson[10], James F Smith[1], Christopher D Pilcher[10], Friedrich Gagsteiger[4], Olena Sakk[11], Michael O'Rand[6§], Polina V Lishko[9], Frank Kirchhoff[3], Jan Münch[3‡], Warner C Greene[2,8,12‡]

[1]Department or Urology, University of California San Francisco, San Francisco, United States; [2]Gladstone Institute of Virology and Immunology, University of California San Francisco, San Francisco, United States; [3]Institute of Molecular Virology, Ulm University Medical Center, Ulm, Germany; [4]Kinderwunsch-Zentrum, Ulm, Germany; [5]The Center for Immunology and Inflammatory Diseases, Massachusetts General Hospital, Harvard Medical School, Boston, United States; [6]Department of Cell Biology and Physiology, University of North Carolina, Chapel Hill, United States; [7]Core Facility Functional Peptidomics, Ulm University, Ulm, Germany; [8]Department of Medicine, University of California San Francisco, San Francisco, United States; [9]Department of Molecular and Cell Biology, University of California Berkeley, Berkeley, United States; [10]HIV / AIDS Division, San Francisco General Hospital, University of California San Francisco, San Francisco, United States; [11]Core Facility Transgenic Mice, Medical Faculty, Ulm University, Ulm, Germany; [12]Department of Microbiology and Immunology, University of California, San Francisco, United States

*For correspondence: nadia.roan@ucsf.edu

[†]These authors also contributed equally to this work
[‡]These authors contributed equally to this work

Present address: [§]Eppin Pharma Inc., Chapel Hill, United States

Competing interests: The authors declare that no competing interests exist.

**Abstract** Unlike other human biological fluids, semen contains multiple types of amyloid fibrils in the absence of disease. These fibrils enhance HIV infection by promoting viral fusion to cellular targets, but their natural function remained unknown. The similarities shared between HIV fusion to host cell and sperm fusion to oocyte led us to examine whether these fibrils promote fertilization. Surprisingly, the fibrils inhibited fertilization by immobilizing sperm. Interestingly, however, this immobilization facilitated uptake and clearance of sperm by macrophages, which are known to infiltrate the female reproductive tract (FRT) following semen exposure. In the presence of semen fibrils, damaged and apoptotic sperm were more rapidly phagocytosed than healthy ones, suggesting that deposition of semen fibrils in the lower FRT facilitates clearance of poor-quality sperm. Our findings suggest that amyloid fibrils in semen may play a role in reproduction by participating in sperm selection and facilitating the rapid removal of sperm antigens.

## Introduction

Seminal plasma (SP) is a unique biological fluid, harboring unusually high concentrations of proteases and protease inhibitors (*Laflamme and Wolfner, 2013*), immunomodulatory cytokines such as TGF-β (*Robertson et al., 2002*), and metals such as zinc, all of which serve important functions in promoting reproductive success. Thus far, SP is also the only human biological fluid known to contain endogenous amyloid fibrils in a non-disease state (*Usmani et al., 2014*). Two classes of semen

**eLife digest** Seminal plasma, the fluid portion of semen, helps to transport sperm cells to the egg during sexual reproduction. Seminal plasma contains numerous proteins that help the sperm to survive and, in recent years, researchers discovered that it also harbours protein deposits known as amyloid fibrils.

Such protein deposits are generally associated with neurodegenerative diseases such as Alzheimer's and Parkinson's disease, where a build-up of fibrils can damage the nervous system. Semen amyloids, however, are present in the absence of disease, but can boost infection by HIV and other sexually transmitted viruses, by shuttling virus particles to their target cells. Despite these damaging effects, some researchers had suggested that amyloids in semen could be beneficial for humans, though it was unclear what these benefits might be.

Roan et al. now set out to assess how semen amyloids affect human sperm activity. The results show that semen amyloids bind to damaged sperm cells and immobilize them, which are then quickly cleared away by immune cells. This could ensure that only the fittest sperm cells reach the egg.

These findings suggest that amyloids can potentially serve beneficial roles for reproduction. A next step will be to investigate how semen amyloids trap unwanted sperm and how immune cells know when to remove it. More research is needed to investigate if problems in these processes could lead to infertility in men.

amyloids have been identified: those derived from proteolytic fragments of prostatic acid phosphatase (PAP) which polymerize to form amyloids named semen-derived enhancer of viral infection (SEVI), and those derived from PSA-generated fragments of semenogelins which polymerize to form amyloids named SEM fibrils. Both sets of amyloids markedly enhance HIV infection by electrostatically binding HIV virions and increasing their propensity to bind to and infect cellular targets (*Münch et al., 2007*; *Kim et al., 2010*; *Roan et al., 2011*; *Arnold et al., 2012*; *Roan et al., 2014*). Because the ability of semen and SP to enhance HIV infection directly correlates with endogenous levels of these fibrils (*Kim et al., 2010*; *Roan et al., 2014*), inhibiting the activity of semen amyloids may decrease HIV transmission rates. Indeed, semen fibrils are currently being pursued as targets for HIV microbicide development (*Olsen et al., 2010*; *Roan et al., 2010*; *Hartjen et al., 2012*; *Lump et al., 2015*).

While the effects of SP amyloids on HIV infection have been extensively studied, the normal physiological function of these fibrils is unclear. SEM proteins have undergone extensive positive selection over evolutionary time (*Hurle et al., 2007*; *Ferreira et al., 2013*), suggesting an important role for these proteins in evolutionary fitness. Studies analyzing orthologs of the human amyloidogenic SEM peptide from 12 non-human primate species revealed that the amyloidogenic potential of these orthologous peptides and their virus-enhancing properties are conserved amongst great apes (*Roan et al., 2014*). Whether this selective pressure occurs at the level of the fibrils or the parent protein is not known, but the existence of amyloidogenic semen peptides from multiple primate species suggest that these structures may serve a physiological function. Furthermore, the observation that SP amyloids promote infection by multiple sexually transmitted viruses (*Tang et al., 2013*; *Torres et al., 2015*) suggests that they should be selected against during primate evolution unless they serve a significant physiological purpose. Here, we examined the effects of semen amyloid fibrils on sperm function, and show that they participate in sperm selection and disposal.

## Results

### Semen fibrils inhibit fusion of spermatozoa to oocytes

Because parallels exist between the fusion of HIV to cells and the fusion of sperm to oocyte (*Doncel, 2006*), we first examined whether semen fibrils promote fertilization. Because endogenous semen fibrils behave similarly to synthetic versions of the fibrils and are difficult to isolate as purified materials (*Roan et al., 2014*; *Usmani et al., 2014*), we used fibrils derived from synthetic peptides

for the majority of our studies. Synthetic SEVI and SEM peptides were confirmed to form fibrils by thioflavin T (ThT) staining and electron microscopy (*Figure 1—figure supplement 1*). These synthetic fibrils, like their endogenous counterparts, include both fibrils and fibrillar oligomers as well as prefibrillar oligomers, as determined by their reactivity with amyloid conformer-specific antibodies OC and A11 (*Figure 1—figure supplement 2*). In the remainder of this manuscript, we use the term 'fibrils' to refer to the synthetic form of the amyloids, and 'endogenous amyloids' when fibrils were purified from semen.

For ethical reasons, we conducted in vitro fertilization (IVF) using mouse instead of human gametes. Given that the infection-promoting effects of the fibrils are driven by electrostatic forces and is not receptor-specific (*Roan et al., 2009*, *2011*), if fertilization is enhanced by the fibrils, then the effect should be species-independent. Contrary to our hypothesis, both SEVI and SEM fibrils decreased IVF rates in a dose-dependent manner (*Figure 1*). Significant inhibition of IVF was observed at fibril concentrations of 50 µg/ml, and near complete inhibition was achieved with 250 µg/ml. Because concentrations of amyloidogenic peptides in semen range from 28 to 267 µg/ml (*Münch et al., 2007*; *Roan et al., 2011*, *2014*), these results suggest that physiologically relevant concentrations of semen fibrils suppress IVF. This inhibition was not due to fibril-induced cytotoxicity to the spermatozoa, oocytes, or embryos, as evidenced by propidium iodide (PI) staining experiments (data not shown).

## Semen fibrils immobilize spermatozoa

To clarify the mechanism underlying reduced IVF rates in the presence of the fibrils, we performed live cell imaging. We found that the fibrils seem to inhibit fusion of sperm to oocyte by entrapping mouse sperm cells (*Figure 1—figure supplement 3*; *Video 1*). Fibrils similarly entrapped human spermatozoa in a dose-dependent manner, as assessed by both manual quantitation as well as computer-assisted sperm analysis (CASA) (*Figure 2A*, *Figure 2—figure supplement 1*). Close examination of fibril-exposed human spermatozoa by cryosection electron microscopy revealed that the fibrils directly interacted with the plasma membranes of both sperm heads and tails, and that points of contact tended to extend the membrane away from the base of the sperm head and tail (*Figure 2B*). To confirm that endogenous amyloids also associate with sperm, we fractionated human SP pooled from 20 donors and obtained a fraction containing endogenous amyloids as demonstrated by ThT binding (*Figure 2—figure supplement 2*). Microscopic analysis showed that these purified endogenous amyloids associated with spermatozoa, as did endogenous amyloids present in fresh liquefied ejaculates (*Figure 2—figure supplement 3*). Furthermore, purified endogenous amyloids, like synthetic fibrils, efficiently entrapped spermatozoa (*Figure 2C*).

To examine the effects of the fibrils on sperm motility at the single cell level, we assayed the motility of spermatozoa from freshly ejaculated human semen by video microscopy. Under HEPES-buffered conditions, spermatozoa attached by their heads to coverslips exhibited continuous beating motions of the tail (*Video 2A*). Remarkably, within 10 min after perfusion of the SEM fibrils, some spermatozoa became completely immotile, while others exhibited twitching movements (*Video 2B*, *Figure 2D*). In the presence of the fibrils, 89.8 ± 9.3% of spermatozoa (average data from three donors) were fully or partly immobilized as defined by at least 50% of the tail being surface-immobilized. Of note, this immobilization is distinct from the previously described ability of the SEM1 holoprotein to limit sperm motility by binding to the sperm-associated EPPIN complex (*Mitra et al., 2010*), since that activity, mapped to cysteine 239 of SEM1, is not part of the amyloidogenic SEM1 fragment (*Roan et al., 2014*). SEM fibril-induced immobilization did not cause changes in PI uptake, mitochondrial activity, or ability to capacitate relative to control peptide (*Figure 2—figure supplement 4*), showing that immobilized spermatozoa were not compromised in viability due to exposure to amyloid structures. In addition, perfusion of seminal proteases onto fibril-immobilized spermatozoa partially restored sperm motility (*Video 3*) further confirming lack of cytotoxicity. Because proteolytic degradation of amyloidogenic peptides in semen occurs gradually over the course of liquefaction (*Roan et al., 2014*), immobilization of sperm is likely most efficient during the earliest stages post-ejaculation.

We further demonstrated that the ability of SEM fibrils to immobilize spermatozoa was linked to its fibrillar structure by showing that SEM1(68–85), a naturally-occurring SEM-derived peptide in semen that does not form fibrils (*Roan et al., 2011*), did not inhibit motility (*Video 4A*, *Figure 2D*). Motility was also not inhibited by SEM1(108–159), a SEM-derived peptide that is non-fibrillar but

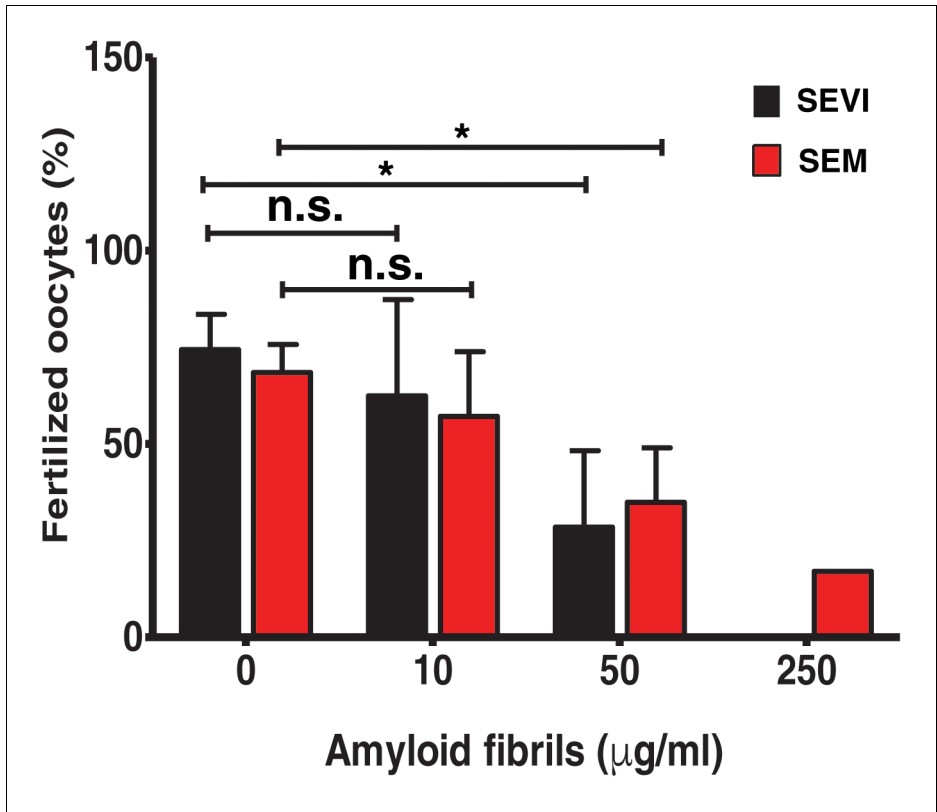

**Figure 1.** SEVI and SEM fibrils inhibit IVF in a dose-dependent manner. The indicated concentrations of SEVI and SEM fibrils were added to mouse spermatozoa and oocytes, and monitored for IVF rates as detailed in the Materials and methods section. *p<0.05 (two-tailed Student's t test). n.s. = non-significant. Error bars reflect variation between different experiments conducted using gametes from different mice, and correspond to data averaged from 3 to 5 experiments. In experiments with SEVI, the number of oocytes fertilized were 197/272 (0 μg/ml SEVI), 112/179 (10 μg/ml SEVI), 67/219 (50 μg/ml SEVI), and 0/77 (250 μg/ml SEVI). In experiments with SEM fibrils, the number of oocytes fertilized were 78/116 (0 μg/ml SEM), 91/162 (10 μg/ml SEM), 39/128 (50 μg/ml SEM), and 6/35 (250 μg/ml SEM). The 250 μg/ml condition lacks error bars as it was only tested in two experiments due to limited cell numbers; in both of these experiments treatment with 250 μg/ml SEVI led to complete abrogation of IVF (0% fertilized oocytes).

The following figure supplements are available for figure 1:

**Figure supplement 1.** Confirmation of fibril formation by SEVI and SEM peptides.

**Figure supplement 2.** Amyloid conformer analysis of seminal plasma and synthetic semen fibrils.

**Figure supplement 3.** Semen fibrils trap mouse spermatozoa and inhibit their progressive motility.

highly cationic (pI = 10.12) (*Roan et al., 2011*) (*Video 4B*, *Figure 2D*). In contrast, human SEVI (*Münch et al., 2007*) and the previously described SEM-derived fibril from the non-human primate *Otolemur garnettii* (referred to as Galago) (*Roan et al., 2014*) both immobilized spermatozoa (*Video 5A,B*, *Figure 2D*), while Aβ(1–42) fibrils, not naturally present in semen, did not cause immobilization (*Video 5C*, *Figure 2D*). All together, these data suggest that semen amyloid fibrils, and not native semen peptides or pathological amyloids, are distinct in their ability to immobilize sperm cells. The differential effects of semen fibrils vs. Aβ(1–42) fibrils on sperm motility could be due to differences in fibril charge and/or distribution of amyloid conformers.

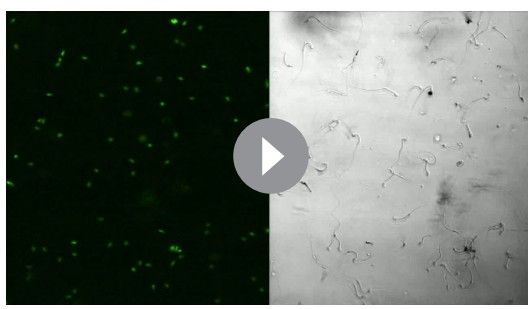

**Video 1.** Semen fibrils entrap mouse spermatozoa. 10[5] mouse spermatozoa stained with Hoechst 33342 (green) were incubated in the absence (A) or presence (B) of 50 µg/ml of semen fibrils at 37°C in a final volume of 100 µl for 15–20 min. Images were acquired for 20 s with an interval of 1 s on an LSM710 confocal microscope (Zeiss) using a 20X air objective.

## Semen fibrils promote phagocytosis of damaged spermatozoa

Having established that semen fibrils do not promote fusion of sperm to egg, but that they do uniquely immobilize sperm cells, we next sought to address what effect this may have on reproduction. Sexual intercourse elicits a massive infiltration of neutrophils and macrophages into the female reproductive tract (FRT) (*Pandya and Cohen, 1985*; *Sharkey et al., 2012*), presumably to mediate clearance of microorganisms and remnant sperm cells, and perhaps to filter out morphologically abnormal and/or non-functional sperm cells (*Tomlinson et al., 1992*; *Oren-Benaroya et al., 2007*). Thus, we next tested whether spermatozoa entrapped by the fibrils are preferentially phagocytosed.

Macrophages were differentiated from human monocytes obtained from female donors, confirmed for phagocytic activity (*Figure 3—figure supplement 1A*), and then imaged by confocal microscopy following incubation with fluorescently-labeled spermatozoa. Macrophages that had taken up multiple sperm cells could be readily observed (*Figure 3—figure supplement 1B*, *Videos 6* and *7*). Higher throughput imaging with the Amnis Imagestream revealed that some macrophages had taken up a single spermatozoon while others had taken up multiple ones (*Figure 3—figure supplement 1C*). Having demonstrated the ability to assess phagocytosis of spermatozoa in vitro, we next used a biochemical method to assess whether fibrils increase this process. Spermatozoa were added to cultured macrophages, and at various timepoints the macrophages were washed extensively to remove surface-associated spermatozoa and cell lysates were prepared for Western blotting. Acetylated tubulin, a ciliary protein expressed at high levels in sperm but not in somatic cells, was used as a marker for phagocytosed sperm cells. As shown in *Figure 3A*, phagocytosis of spermatozoa was apparent by 3 hr, and markedly increased by semen fibrils. Acetylated tubulin was not detected when the assay was performed at 4°C to prevent phagocytic activity (*Figure 3A*), verifying detection of actual phagocytosis as opposed to surface binding of sperm cells.

To quantify spermatozoa uptake by macrophages at the single-cell level, we developed a FACS-based phagocytosis assay. Spermatozoa were fluorescently labeled with eFluor 670 and then co-cultured with macrophages. Cell surface expression of CD14 and CD33 was used to differentiate the macrophages from the sperm cells. Macrophages that had taken up spermatozoa were identified by eFluor 670 fluorescence. As demonstrated in *Figure 3—figure supplement 2A*, a distinct population of macrophages that had phagocytosed spermatozoa was readily apparent after 0.5 hr of co-culture. This population was abrogated when the assay was conducted in the presence of the phagocytosis inhibitor cytochalasin D or when the assay was conducted at 4°C instead of 37°C (*Figure 3—figure supplement 2A*). Thus, detection of sperm-harboring macrophages was not due to leakage of dye from spermatozoa, or from cell-surface binding of the spermatozoa to the macrophages. Furthermore, >99% of the macrophages at the time of harvest were viable (*Figure 3—figure supplement 2B*), excluding non-specific effects due to autofluorescence of dying cells. To assess whether semen fibrils promote phagocytosis of spermatozoa, we conducted the FACS-based phagocytosis assay in the absence and presence of semen fibrils. Consistent with the Western blot data, addition of fibrils increased phagocytosis of spermatozoa, from 4.61% to 22.6% (*Figure 3B*).

Because removal of damaged or defective sperm cells from the reproductive tract may be important to rapidly clear the lower FRT of potentially immunogenic male antigens, we also assessed phagocytosis of damaged spermatozoa. Sperm cells damaged by multiple freeze/thaw cycles in liquid nitrogen or electromagnetic radiation were taken up at higher rates than healthy sperm cells (*Figure 3—figure supplement 3A*). Because liquid nitrogen exerted a more potent effect (up to 3.4-fold increased uptake, as opposed to up to 2.3-fold increased uptake for radiation-damaged

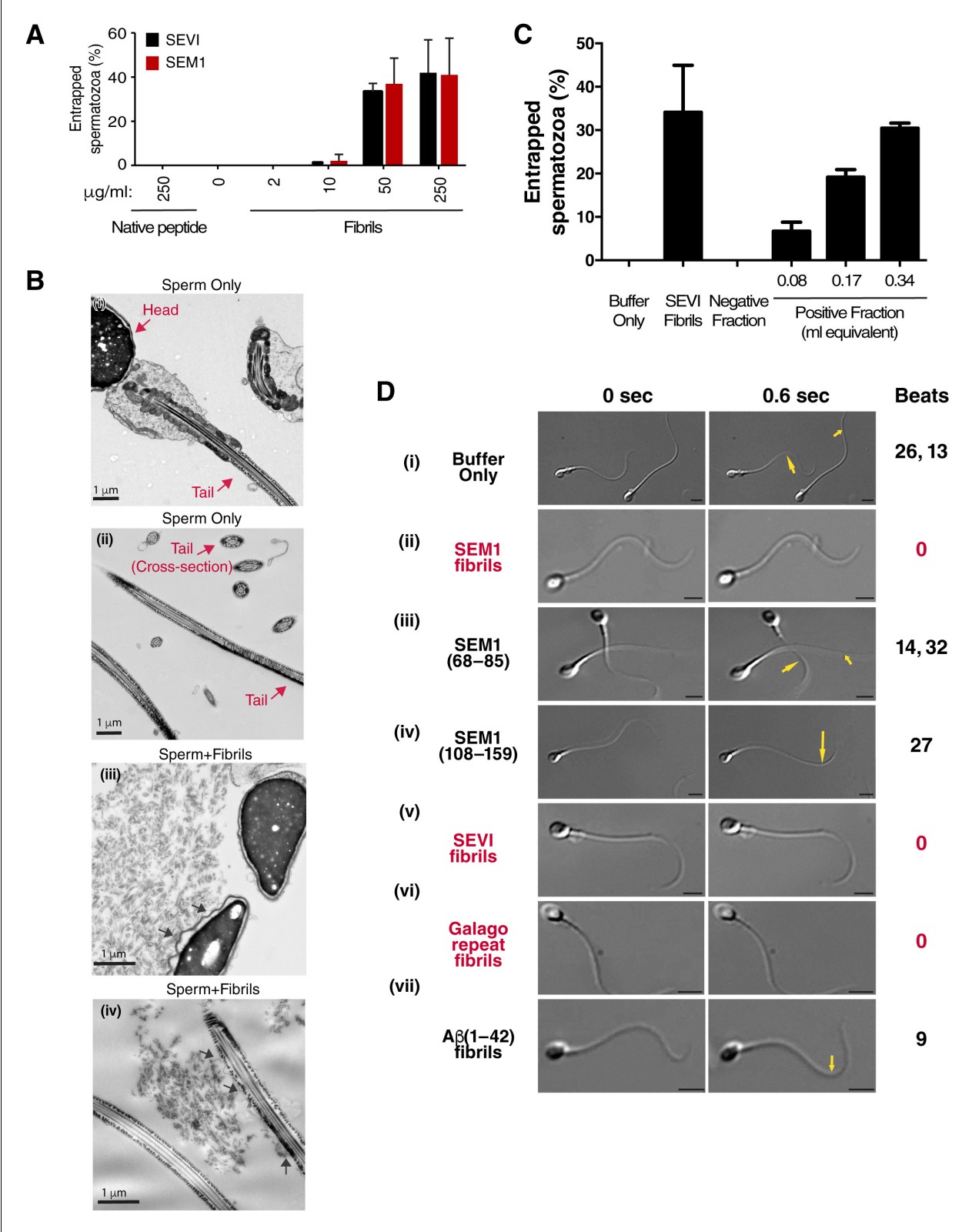

**Figure 2.** Semen fibrils directly bind and immobilize human spermatozoa. (**A**) Human spermatozoa incubated with semen fibrils were imaged at 37°C for 5–10 min and then assessed for % entrapped spermatozoa as described in the Materials and methods section. Native peptide corresponds to monomeric, non-fibrillized peptide. (**B**) Spermatozoa were incubated in the absence (i, ii) or presence (iii, iv) of SEM fibrils and then imaged by sectioning electron microscopy. Image in panel (iii) shows two sperm heads and image in panel (iv) shows two sperm tails, with arrows highlighting
*Figure 2 continued on next page*

*Figure 2 continued*

examples of interactions between the fibrils and the tail. (**C**) Spermatozoa treated with fractions containing (Positive Fraction) or lacking (Negative Fraction) endogenous semen amyloids were imaged for 5–10 min at 37°C and then assessed for % entrapped spermatozoa as described in the Materials and methods section. Treatment of spermatozoa with synthetic SEVI fibrils was used as a positive control for entrapment. The buffer only and negative fraction controls exhibited 0% entrapment. (**D**) Sperm motility was assessed before (i) or after (ii–vii) perfusion with 50 µg/ml of SEM fibrils (ii), SEM1(68–85) (iii), SEM1(108–159) (iv), SEVI (v), the *O. garnettii* (Galago) SEM2 repeat amyloid fibrils (vi), or A$\beta$(1–42) (vii). Within each pair of images, the first corresponds to t = 0, whereas the second corresponds to t = 0.6 s. Numbers correspond to the number of beats that occurred within the length of each movie (total time = 2 s). In instances where two numbers are shown, the first corresponds to the spermatozoon on the left and the second to the spermatozoon on the right. Red text highlights samples where spermatozoa were immobilized. Yellow arrows highlight tail regions within the second frame that moved relative to the first frame. Scale bars = 5 µm. Data for each treatment are representative of at least two independent experiments examining >5 individual spermatozoa per treatment.

The following figure supplements are available for figure 2:

**Figure supplement 1.** Semen fibrils immobilize spermatozoa in CASA.

**Figure supplement 2.** Confirmation of fibrillar nature of purified endogenous amyloids by thioflavin T.

**Figure supplement 3.** Spermatozoa interact with endogenous amyloids in human semen.

**Figure supplement 4.** Fibril-immobilized spermatozoa are metabolically active and viable.

---

spermatozoa), we selected this method of damage induction for subsequent experiments. In the absence of the fibrils, the rate of phagocytosis of liquid nitrogen-damaged spermatozoa was higher than that of fresh spermatozoa (31.3% versus 4.61%, respectively); this was also observed in the presence of the fibrils but at higher percentages (49.9% versus 22.6%, respectively) (*Figure 3B*). Importantly, the absolute numbers of spermatozoa added to each co-culture condition were the same to allow direct comparisons between conditions. Interestingly, with longer incubation times high levels of phagocytosis of both healthy and damaged sperm cells were achieved in the presence of fibrils, suggesting that the fibrils promote the kinetics of sperm phagocytosis (*Figure 3—figure supplement 3B*). Confocal microscopy revealed that fibrils also increased the number of spermatozoa engulfed by individual macrophages, frequently resulting in a single macrophage endocytosing more than a dozen sperm heads (*Figure 3—figure supplement 4*, *Videos 8* and *9*).

We consistently observed in multiple donors that in the presence of the fibrils damaged spermatozoa are preferentially phagocytosed over healthy ones (*Figure 3B*, *Figure 3—figure supplement 3A*). To confirm this phenomenon, healthy and damaged spermatozoa were co-cultured with macrophages in the absence or presence of SEM fibrils, both at 37°C as well as 4°C to block phagocytic activity. As shown in *Figure 3C*, the highest level of phagocytosis was observed with damaged spermatozoa in the presence of the fibrils, in line with our prior experiments. However, macrophage/sperm samples incubated at 4°C with SEM fibrils also exhibited positive events within the phagocytosis gate, suggesting that SEM caused some level of sperm sticking to the macrophage surface (*Figure 3C*). To normalize for the contribution of cell-surface sticking, we subtracted out the contribution of the positive events at 4°C for each sample. Levels of phagocytosis of damaged spermatozoa in the presence of fibrils remained significantly higher than that of healthy

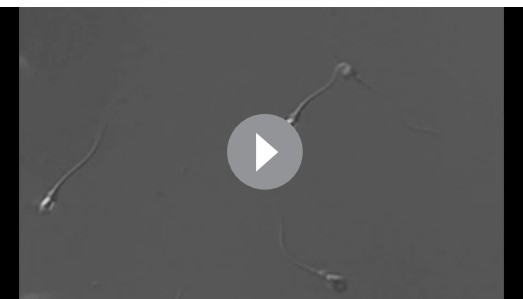

**Video 2.** Spermatozoa from fresh ejaculates are immobilized by SEM fibrils. (A) Spermatozoa were isolated as detailed in the supplemental experimental procedures, attached onto coverslips, and examined for motility by live microscopy. (B) Cells were then perfused with 50 µg/ml SEM1(86–107) fibrils and motility was assessed by video microscopy after 10 min. Of note, the same four spermatozoa are shown in the two panels.

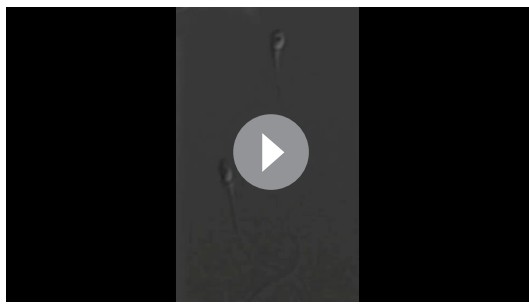

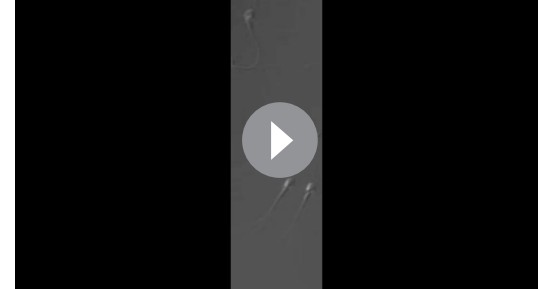

**Video 3.** Immobilization of surface-associated sperm cells by SEM1(86–107) fibrils is reversible. Motile spermatozoa were examined before (A) and 10 min after (B) perfusion with SEM1(86–107) amyloid fibrils. 0.45 µm-filtered seminal plasma was then perfused in at a concentration of 20% as a source of seminal proteases, and 20 min later sperm cells were assessed for motility (C). Data are representative of n = 6 experiments from two sperm donors.

**Video 4.** PSA-generated non-fibrillar fragments SEM1 (68–85) and SEM1(108–159) do not inhibit sperm motility. Motile spermatozoa were examined for motility before and 10 min after perfusion with 50 µg/ml SEM1(68–85) (A) or SEM1(108–159) (B).

spermatozoa under these conditions (*Figure 3D*). All together, these results suggest that: (1) macrophages preferentially phagocytose damaged spermatozoa both in the absence and presence of semen fibrils, consistent with prior reports that increased levels of macrophages are associated with decreased levels of abnormal spermatozoa in semen (*Tomlinson et al., 1992*), and (2) within the first hour, the highest levels of damaged sperm phagocytosis are observed in the presence of semen fibrils.

To directly assess the preference for phagocytosis of damaged spermatozoa, we established a competition assay enabling visualization of normal and damaged spermatozoa within the same well by labeling the two populations with different fluorescent dyes. Notably, macrophages that had taken up only damaged spermatozoa or healthy spermatozoa as well as combinations of both could be detected (*Figure 3—figure supplement 5*). We found that although damaged spermatozoa were preferentially phagocytosed both in the absence and presence of the fibrils, the highest rate of damaged sperm phagocytosis occurred in the presence of fibrils, when 36.8% of macrophages contained damaged spermatozoa (*Figure 3—figure supplement 5*). Of these, 26.3% had only damaged spermatozoa, while 10.5% had both healthy and damaged spermatozoa. These results confirm that damaged sperm cells are efficiently phagocytosed in the presence of semen fibrils.

## Semen fibrils efficiently enhance phagocytosis of apoptotic spermatozoa

Spermatozoa are highly prone to undergo apoptosis, and this process is accelerated by reactive oxygen species produced by surrounding leukocytes as well as the sperm cells themselves (*Aitken et al., 2012*). One of the hallmarks of apoptotic sperm cells is externalization of phosphatidylserine, a negatively charged phospholipid membrane component. Because all known semen fibrils are highly cationic and bind to anionic membrane components (*Roan et al., 2009*, *2011*), we reasoned they may also entrap and promote phagocytosis of apoptotic sperm cells. To test this, we incubated swim-up sperm cells for 24 hr at 25°C. Because swim-up sperm cells are no longer in the antioxidant

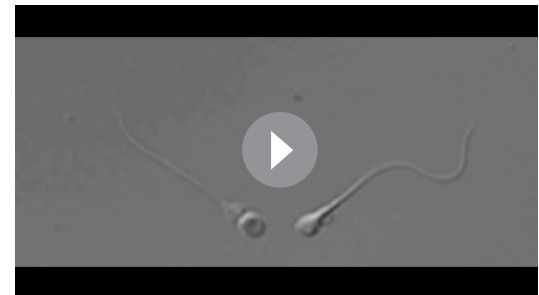

**Video 5.** SEVI and *O. garnettii* SEM2 repeat fibrils inhibit sperm motility, whereas Aβ(1–42) fibrils do not. Motile spermatozoa were examined for motility before and 10 min after perfusion with 50 µg/ml SEVI fibrils (A), *O. garnettii* (Galago) SEM2 repeat fibrils (B), or Aβ (1–42) fibrils (C). An Aβ(1–42) fibril concentration of 100 µg/ml, corresponding to an equimolar amount of 50 µg/ml SEM1(86–107), was also tested and did not inhibit sperm motility (data not shown).

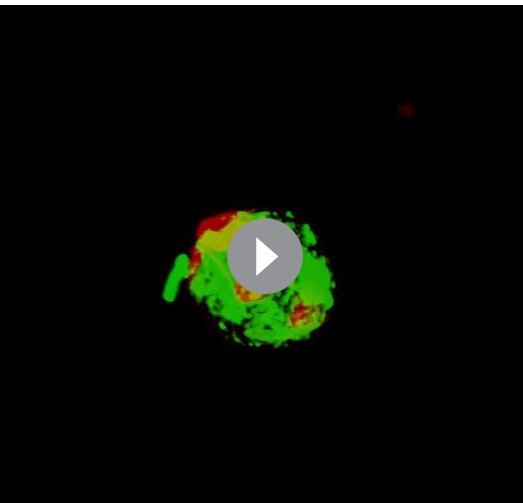

**Video 6.** Internalization of spermatozoa by macrophage (rotational view). Monocyte-derived macrophages labeled with a membrane dye (green) were incubated with sperm cells (red) for 3 hr and then imaged by confocal microscopy.

**Video 7.** Internalization of spermatozoa by macrophage (z-stacks view). Monocyte-derived macrophages labeled with a membrane dye (green) were incubated with sperm cells (red) for 3 hr and then imaged by confocal microscopy.

environment of seminal plasma, they are prone to undergo their default cascade of apoptotic cell death (*Aitken et al., 2012*). Annexin V staining confirmed that compared to fresh sperm cells, a higher proportion of the 24 hour-treated sperm cells were apoptotic (*Figure 4A,B*). The fresh vs. treated sperm cells were then added to macrophages in the absence or presence of fibrils, and phagocytosis was monitored. Like damaged spermatozoa, the highest levels of phagocytosis of apoptotic sperm cells was observed under conditions where semen fibrils were present (*Figure 4C*).

## Discussion

In conclusion, we demonstrate that semen fibrils promote phagocytosis of sperm cells by macrophages. Because the highest rates of damaged sperm phagocytosis occurred in the presence of semen fibrils, the fibrils may participate in quality control by promoting the efficiency of sperm selection by macrophages. Moreover, by also promoting phagocytosis of non-damaged sperm, the fibrils appear to additionally participate in rapidly clearing the lower reproductive tract of remaining sperm cells (whether damaged or not), which may help avoid the development of an inappropriate cell-mediated immune response against sperm antigens. The rapid removal of sperm cells is consistent with the observation that the lower reproductive tract largely returns to its pre-mating state by 24 hr post-coitus (*Pandya and Cohen, 1985*), and may help explain why only a minute fraction of deposited spermatozoa actually reach the oviduct isthmus (*Hunter, 1993*; *Eisenbach and Giojalas, 2006*). This in turn may be important for fertilization and blastocyst development since high concentrations of spermatozoa can result in increased polyploidy rates and be detrimental for human embryonic development (*Mahadevan and Trounson, 1984*; *Diamond et al., 1985*; *Englert et al., 1986*; *Dumoulin et al., 1992*). Future studies should assess the molecular basis by which semen fibrils entrap sperm and promote their elimination, including the nature of the interaction between the fibrils and the sperm membrane. In addition, because semen amyloids are structurally diverse (*Usmani et al., 2014*) and include both fibrillar and pre-fibrillar oligomers (*Figure 1—figure supplement 2*), further studies to determine whether specific conformers are responsible for sperm binding and entrapment are warranted.

Of note, semen fibrils may not be the only structures in semen that can participate in sperm disposal; neutrophil extracellular traps (NETs) – chromatin- and protein-containing extracellular fibers extruded by neutrophils to trap and kill bacteria (*Brinkmann et al., 2004*) – have been shown to

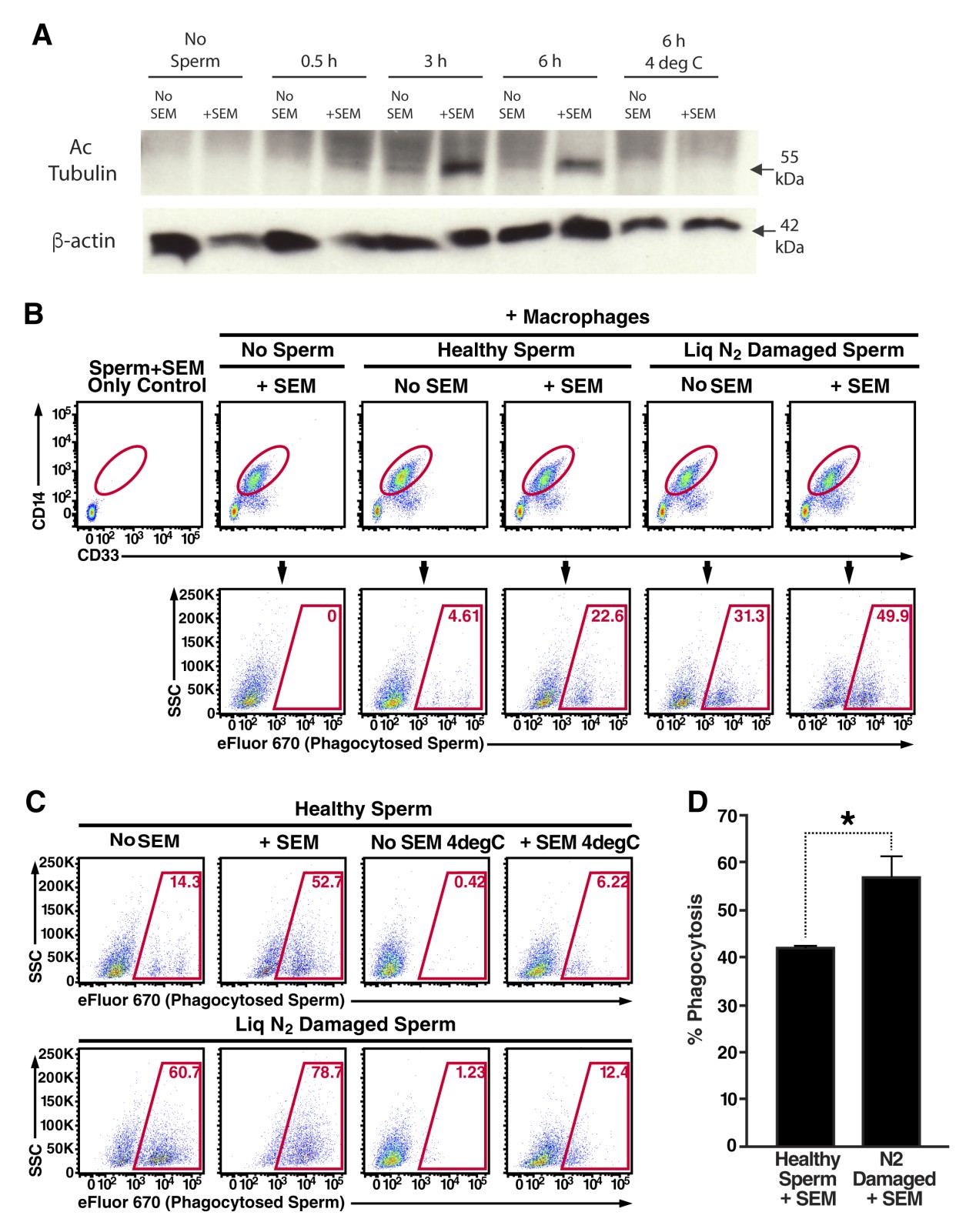

**Figure 3.** Semen fibrils promote phagocytosis of sperm cells. (**A**) Spermatozoa were added to monocyte-derived macrophages for the indicated number of hours in the presence or absence of 100 µg/ml SEM fibrils, washed, and then blotted for acetylated tubulin (to detect spermatozoa) or *β*-actin (to detect macrophages). Negative controls include macrophages in the absence of spermatozoa, and incubation of macrophages with spermatozoa at 4°C to prevent phagocytosis. (**B**) Motile sperm cells purified by the swim-up method were labeled with eFluor 670 and then left at room

*Figure 3 continued on next page*

*Figure 3 continued*

temperature or damaged by five sequential rounds of freeze-thaw with liquid nitrogen. Spermatozoa were then added to monocyte-derived macrophages for 0.5 hr at 37°C in the presence or absence of 100 μg/ml SEM fibrils, washed, and then assessed by flow cytometry. Macrophages were identified by gating on CD14+CD33+ cells, and phagocytosis was assessed by determining the percentages of macrophages that were eFluor 670+. Results are representative of data from five different donors. (C) Comparison of eFluor 670+ macrophages after incubation with labeled spermatozoa at 37°C vs. 4°C (temperature at which phagocytosis is inhibited). These data suggest that in the presence of SEM fibrils, a small number of macrophages have surface-associated spermatozoa. (D) Macrophage-mediated phagocytosis of healthy vs. damaged spermatozoa in the presence of SEM fibrils was compared in triplicates. Shown values are those where the levels of eFluor 670+ macrophages in the presence of the SEM fibrils under 4°C conditions were subtracted out. This normalization was performed to discount surface-associated spermatozoa (which is present to some extent as demonstrated in panel C) from the analysis. *$p<0.05$ (by 2-tailed $t$ test).

The following figure supplements are available for figure 3:

**Figure supplement 1.** Internalization of spermatozoa by macrophages.

**Figure supplement 2.** Macrophage-mediated phagocytosis of sperm cells can be detected by flow cytometry.

**Figure supplement 3.** Effect of damage induction and fibrils on sperm phagocytosis.

**Figure supplement 4.** High number of spermatozoa internalized by a single macrophage in the presence of semen fibrils.

**Figure supplement 5.** Semen fibrils increase the percentage of macrophages that have phagocytosed both healthy and damaged spermatozoa.

entrap spermatozoa (*Alghamdi and Foster, 2005*), which could conceivably promote their subsequent phagocytic uptake. Importantly, however, NETs are different from the semen fibrils because they inhibit rather than enhance HIV infection (*Saitoh et al., 2012*). As such, while semen fibrils have been proposed as targets for HIV microbicide development (*Olsen et al., 2010*; *Roan et al., 2010*; *Hartjen et al., 2012*; *Roan and Münch, 2015*), this would not apply to NETs. Should semen fibril antagonists advance to clinical stages, one should be cautious as to potential effects on sperm cells and fertilization, given the data presented herein.

Amyloid fibrils have long been considered a result of aberrant protein folding that is almost exclusively associated with systemic or localized amyloidosis, or various chronic degenerative diseases (*Chiti and Dobson, 2006*). However, it is becoming increasingly clear that fibrils may also form for beneficial reasons (*Kelly and Balch, 2003*; *Hou et al., 2011*; *Bergman et al., 2016*). Two prominent examples of functional amyloid in humans are HD6 (human α-defensin 6) forming amyloid nets to protect host cells from invasion by enteric bacterial pathogens (*Chu et al., 2012*), and Pmel17 fibrils facilitating the formation of melanosomes with covalently linked melanin (*Fowler et al., 2006*). Prior studies have also suggested that amyloids generated by the male reproductive tract may play beneficial roles in bacterial clearance and sperm maturation (*Whelly et al., 2012*; *Easterhoff et al., 2013*). The present findings add SEVI and SEM fibrils to the growing list of functional amyloids in humans by demonstrating that these fibrils can play an active role in defective sperm disposal.

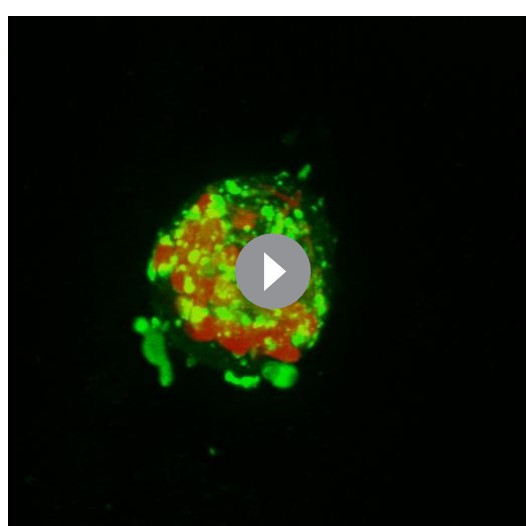

**Video 8.** High number of spermatozoa internalized by a single macrophage in the presence of semen fibrils (rotational view). Monocyte-derived macrophages labeled with a membrane dye (green) were incubated with sperm cells (red) for 3 hr in the presence of 100 μg/ml SEM fibrils and then imaged by confocal microscopy.

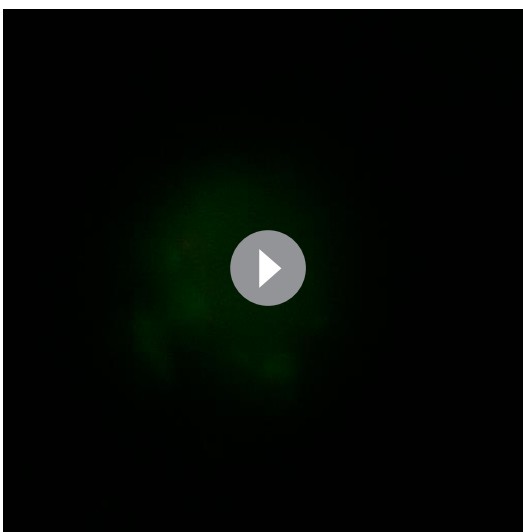

**Video 9.** High number of spermatozoa internalized by a single macrophage in the presence of semen fibrils (z-stacks view). Monocyte-derived macrophages labeled with a membrane dye (green) were incubated with sperm cells (red) for 3 hr in the presence of 100 µg/ml SEM fibrils and then imaged by confocal microscopy.

## Materials and methods

### Peptides and fibrils

Sequences of the semen-derived peptides to generate the SEVI and SEM fibrils have been previously described (*Münch et al., 2007*; *Roan et al., 2011*, *2014*). These peptides were chemically synthesized by Celtek Peptides (Nashville, TN), CPC Scientific (Sunnyvale, CA), or U-PEP (Ulm, Germany), and dissolved in PBS (pH 7.0) at a concentration of 2.5 mg/ml. To accelerate nucleation of fibril formation (*Giehm and Otzen, 2010*), all peptide samples were agitated overnight in PBS at 37°C at 1400 rpm in an Eppendorf Thermomixer (Hauppauge, NY). Synthetic $A\beta(1–42)$ peptides lyophilized from hexafluoroisopropanol (HFIP) solution were purchased from rPeptide (Bogart, GA), diluted to 100 µg/ml, and amyloid formation was promoted by agitating peptides at 1400 rpm for 2 days at 37°C. Amyloid formation was confirmed by both electron microscopy and Thioflavin-T (ThT) analysis as described (*Roan et al., 2014*).

### Dot blot analysis

Nitrocellulose (Hybond ECL, GE Healthcare, Chicago, IL) was pre-wetted with PBS-T (PBS containing 0.1% Tween 20) and then spotted under vacuum with 50 µl 10% seminal plasma, 10% blood plasma, or 20 µg/ml semen amyloids (SEVI or SEM1(86–107) fibrils) or the corresponding monomeric peptides. Wells were then washed once with PBS-T under vacuum. The nitrocellulose membrane was then immediately transferred to a blocking solution (PBS-T containing 5% non-fat dry milk and 1% BSA) and incubated at 4°C with gentle shaking for 2 hr. The membrane was then washed three times for 5 min before incubation with anti-Amyloid Fibrils OC Antibody (EMD Millipore, Billerica, MA) or Anti-Amyloid Oligomers A11 antibody (Abcam, Cambridge, United Kingdom) (both used at 1:2500 in PBS-T with 1% BSA). Primary antibody incubation was allowed to proceed overnight at 4°C with gentle shaking. The membrane was then washed three times for 5 min each before addition of HRP-conjugated rabbit IgG (GE Healthcare) (used at 1:5000 in PBS-T with 1% BSA). Secondary antibody incubation was allowed to proceed for 2 hr at 4°C with gentle shaking. The membrane was then washed three times for 10 min each, and then developed by 1 min incubation with the Western Lightning ECL Pro (Perkin Elmer, Waltham, MA), and exposed using chemiluminescence film (Amersham Hyperfilm ECL, GE Healthcare).

### Mouse sperm isolation

C57Bl/6N or NMRI mice were purchased from JANVIER SAS (Le Genest Saint Isle, France) and bred in-house at the TVZ University of Ulm. All animals were allowed to adjust to the facility for at least one week before they were used in experiments. The animals were maintained in separate microventilated cages (Sealsafe Next IVC Blue Line, Tecniplast, Buguggiate, Italy), with a 12 hr light-dark cycle and food and water *ad libitum*. The use of animals was approved by the Regierungspräsidium Tübingen, registration number 0.185, and was in accordance with existing regulations of the German Federal Law on Care and Use of Laboratory Animals. Mouse spermatozoa were collected directly from the *vas deferens* and *cauda epididymis* of euthanized C57Bl/6N mice. For the entrapment assay, spermatozoa were allowed to swim out of the epididymis, squeezed out of the vas deferens, and then incubated for 1 hr at 37°C to allow for capacitation. Capacitation media consisted of Human Tubal Fluid (HTF) medium (Irvine Scientific, Newtownmountkennedy, Ireland) supplemented with 15 mg/ml BSA (Fraction V, Carl Roth GmbH, Karlsruhe, Germany). For cryopreservation of spermatozoa

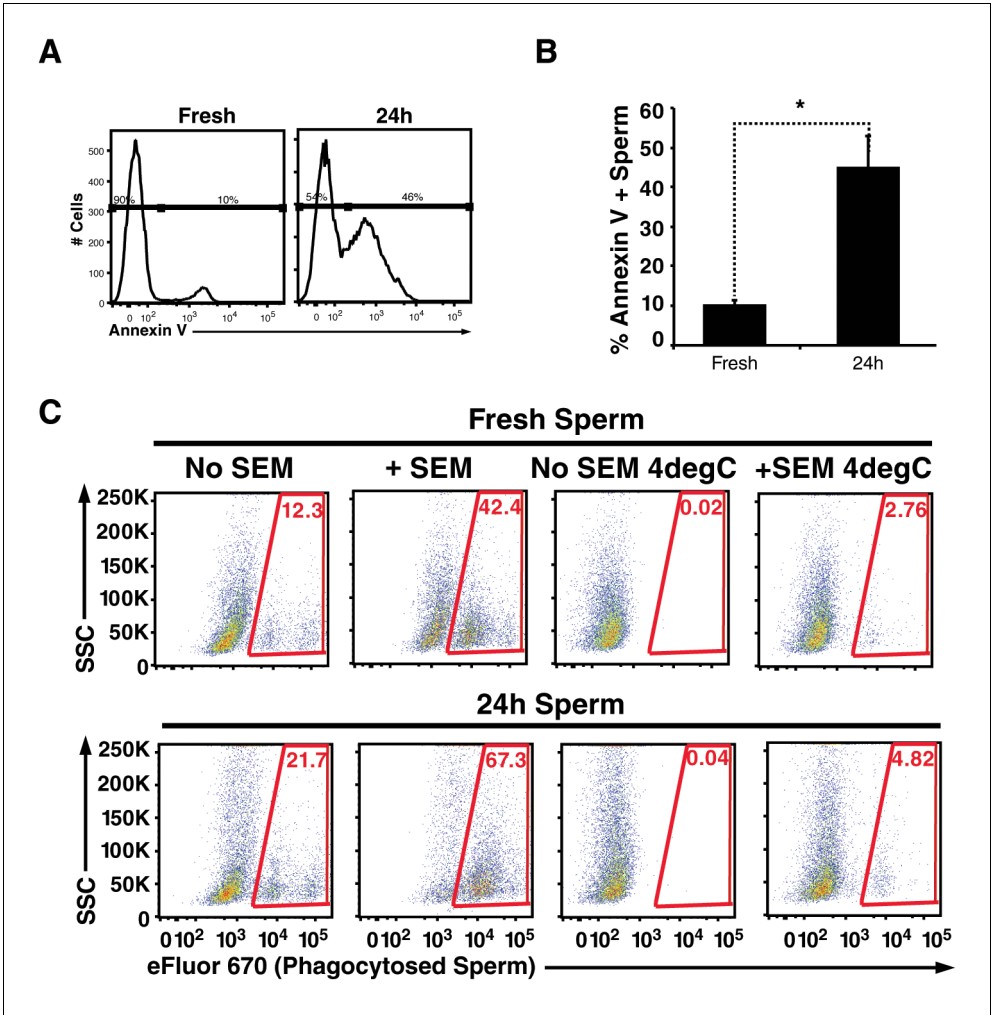

**Figure 4.** Apoptotic sperm cells are efficiently phagocytosed in the presence of fibrils. (**A**) Incubation of spermatozoa for 24 hr at 25°C increases the proportion of apoptotic sperm cells. Spermatozoa from fresh ejaculates were purified by the swim-up technique and then assessed immediately for cell surface expression of the apoptotic marker Annexin V by flow cytometry, or incubated for 24 hr at 25°C prior to staining and analysis. Results are representative of 3 independent donors. Presented are flow cytometric plots showing the percentages of Annexin-negative (non-apoptotic) and Annexin-positive (apoptotic) spermatozoa as indicated. (**B**) Results from experimental triplicates of each condition described in panel **A**. *p<0.05 (two-tailed Student's t test). Results are representative of 3 independent donors. (**C**) Phagocytosis of fresh spermatozoa or spermatozoa incubated for 24 hr at 25°C. Motile spermatozoa purified by the swim-up method were labeled with eFluor 670 and then fed immediately to macrophages or incubated for 24 hr at 25°C to induce apoptosis prior to incubation with macrophages. Assays were conducted in the presence or absence of 100 µg/ml SEM fibrils, and in all cases phagocytosis was allowed to proceed for 0.5 hr prior to flow cytometric analysis. Macrophages were identified by gating on CD14+CD33+ cells, and phagocytosis was assessed by determining the percentages of macrophages that were eFluor 670+. Results are representative of data from three different donors.

for subsequent IVF assay, spermatozoa of NMRI mice with proven fertility were equilibrated for 10 min in cryoprotectant media (18% raffinose and 3% skim milk), aliquoted in cryovials, and placed in the vapor phase of liquid nitrogen for 10 min. The cryovials were then flash frozen with liquid nitrogen and stored in the liquid phase until further use.

## In vitro fertilization (IVF) assay

Female 5–7 week old C57Bl/6N mice were administered intraperitoneally with 0.1 ml containing 5 IU of Pregnant Mare Serum Gonadotropin (PMSG, Intervet, Germany), followed by 0.1 ml containing 5 IU of human chorionic gonadotropin (hCG, Intervet) 47–48 hr later. Cryopreserved mouse NMRI spermatozoa were rapidly thawed 13 hr post-hCG and added to tissue culture dishes at a concentration of $4.8 \times 10^4$ cells/ml in the presence of the indicated concentrations of fibrils. Females were euthanized by cervical dislocation and cumulus-oocyte complexes (COCs) were isolated by tearing up the swollen ampulla in pre-warmed M2 medium (Sigma-Aldrich, St. Louis, MO). Following a 6 hr incubation, COCs were washed in HTF medium and added to the fertilization dishes in a final volume of 0.5 ml. All dishes were covered with mineral oil and allowed to equilibrate for at least 6 hr at 37°C in a humidified atmosphere and 5% $CO_2$ in air. Zygotes were extensively washed in M16 media (Sigma-Aldrich) and further cultured until fertilization assessment the next day. Fertilization rates were determined by counting the total number of 2 cell embryos 22–24 hr post-IVF.

## Imaging interaction of mouse spermatozoa with fibrils

Freshly isolated mouse spermatozoa ($10^7$/ml in 100 µl) were stained with the nuclear stain Hoechst 33342 (shown in green) and incubated with SEVI amyloid fibrils stained with the Proteostat Amyloid Plaque Detection Kit (Enzo Life Sciences, Farmindale, NY) (shown in red) in HTF medium for 15–20 min at 37°C. Proteostat has been used for detection of a diverse array of amyloid fibrils (*Wang et al., 2012*; *Navarro and Ventura, 2014*; *Navarro et al., 2016*; *Taglialegna et al., 2016*) including those from human semen (*Arnold et al., 2012*; *Usmani et al., 2014*; *Lump et al., 2015*). Images were acquired on an LSM710 confocal microscope for 20 s with an interval of 1 s using a 20X air objective. Results are representative of a total of 3–5 experiments performed for each animal (n = 2).

## Human sperm isolation

De-identified fresh human semen samples were obtained the from University of California, San Francisco (UCSF) / San Francisco General Hospital (SFGH) Positive Health Program Research Group (The Options Project, IRB # 10–00301) and the *Kinderwunsch-Zentrum* (Ulm, Germany, Approval # 351/10 and 156/13) after giving informed consent. For single-cell motility assays, confocal microscopy, Western blotting, and flow cytometry, sperm cells were isolated via the sperm swim-up assay similar to methods previously described (*Lishko et al., 2011*). Briefly, 2 ml of semen liquefied at room temperature was carefully underlaid into a 50 ml conical tube containing 8 ml of in-house generated HTF (consisting of 97.8 mM NaCl, 5 mM KCl, 0.2 mM MgSO4, 0.37 mM KH2PO4, 2 mM CaCl2, 20 mM HEPES, 20 mM lactic acid, 0.4 mM Na-Pyruvate, and 3 mM glucose) and incubated for 1 hr at 37°C. The HTF buffer containing motile spermatozoa was then collected without disturbing the solution of semen underneath. For population analyses (entrapment assays), spermatozoa were instead purified by density gradient centrifugation using the PureCeption kit (KB Biosystem Deutschland, ART-2004, Germany).

## Electron microscopy

To confirm fibrillation of synthetic amyloids, samples (500 µg/ml) were prepared with parlodion-filmed carbon-coated grids and 2% potassium phosphotungstate, pH6.5, and analyzed and photographed as previously described (*Roan et al., 2009*). To visualize the physical interaction between sperm cells and the fibrils, swim-up spermatozoa isolated as described above were incubated in the absence or presence of 100 µg/ml SEM fibrils for 1 hr, and then fixed in a 0.1 M sodium cacodylate buffer solution (pH 7.4) containing 2% glutaraldehyde. The samples were then loaded into 200 µm diameter cellulose capillary tubes (Leica Microsystems Inc., Buffalo Grove, IL), post-fixed in 2% osmium tetroxide in the same buffer, stained en block with 2% aqueous uranyl acetate, dehydrated in acetone, infiltrated, and embedded in LX-112 resin (Ladd Research Industries, Burlington, VT). Samples were ultrathin-sectioned on a Reichert Ultracut S ultramicrotome and counter-stained with 0.8% lead citrate. Grids were examined on a JEOL JEM-1230 transmission electron microscope (JEOL USA, Inc., Peabody, MA) and photographed with the Gatan Ultrascan 1000 digital camera (Gatan Inc., Warrendale, PA).

## Human sperm entrapment assay

Swim-up sperm cells were stained with the nuclear dye Hoechst 33342 while amyloid fibrils were stained with the amyloid-binding dye Proteostat or pFTAA (a luminescent-conjugated oligothiophene dye) as previously described (*Usmani et al., 2014*). A total of $10^5$ spermatozoa (final concentration $10^7$/ml) were mixed with 0, 2, 10, 50 and 250 µg/ml of the indicated synthetic amyloid fibril or purified endogenous amyloids (from the equivalent of 0.34, 0.17, or 0.8 ml SP, see details below) in a final volume of 50–100 µl, and then incubated at 37°C for 15–20 min. Images were acquired for 20 s with an interval of 1 s using a Plan-Apochromat 63X/1.40 oil objective lens on a LSM710 confocal microscope (Zeiss, Germany) equipped with Zen-Software (Zeiss). All experiments were performed at 37°C. A total of 3–5 experiments was performed for each donor to calculate the number of entrapped spermatozoa. Only live and motile spermatozoa were considered for analysis. The total number of free spermatozoa and spermatozoa trapped within the amyloid network was determined from all frames over a 20 s period and then the percentage of entrapped cells was calculated.

## Purification of endogenous semen amyloids

Fresh semen samples from 20 donors were allowed to liquefy and then immediately frozen at −20°C. To isolate fractions enriched for amyloids, all samples were thawed simultaneously, pooled (constituting ~26 ml), and processed as described (*Münch et al., 2007*). Briefly, samples were centrifuged (17,000 x g) to separate spermatozoa and SP. The SP was then diluted with 25 ml extraction buffer (1 M acetic acid, 20 mM ascorbic acid, 1 mM EDTA, 2 M sodium chloride, pH 2.0) and stirred for 10 min at 4°C. The extract was then diluted to a volume of 1.5 L with ultrafiltration buffer (0.1 M acetic acid, 20 mM ascorbic acid, 1 mM EDTA, pH 3.0), and then ultrafiltered using a 30 kDa polysulfon membrane (Pall, Quattro Flow Fluid Systems, Pall GmbH, Dreiich, Germany). The filtrate was diluted with water to a volume of 2 L and then applied to a conditioned cation exchange column (Fractogel TSK SP (S) Merck, Darmstadt, Germany, column size: 2 × 12.5 cm). After loading, the column was washed with two column volumes of water at pH 2.5, and eluted with the following buffers: (A) 0.1 M $Na_2HPO_4$, pH 7.4, volume 150 ml, (B) 1 M ammonium acetate, pH 7.0, volume 180 ml, (C) water pH 7.0, volume 180 ml, (D) 0.1 M NaOH, pH 13, volume 200 ml. The eluates B-D were pooled, the pH was adjusted to 3.0 and the pooled eluates were applied to a reverse phase column (Source RPC polystyrol, particle size15 µm, column size 1 × 12.5 cm, Pharmacia, Freiburg, Germany). Bound peptides were eluted using a linear gradient from 95% A (water and 0.1% [v/v] TFA) to 60% B (80% [v/v] acetonitrile and 0.1% [v/v] TFA) in 55 min, from 60% B to 100% B in 10 min using a flow rate of 1.5 ml/min. Protein elution was monitored with an absorbance detector at 214 nm. A total of 40 fractions was collected, freeze-dried and used for ThT analysis to identify a fraction containing amyloids. The presence of amyloid was further verified by microscopy, and staining with the amyloid-binding dyes Proteostat and pFTAA. After all the purification steps, ~60% of the total material was recovered, accounting for ~15 ml of semen. Because the entrapment assays used 2.3% of the starting material and 3-fold dilutions thereof, this corresponded to semen volume equivalents of 0.34 ml, 0.17 ml, and 0.08 ml.

## Thioflavin T fluorescence

To demonstrate the presence of amyloids by thioflavin T, 10 µl synthetic fibrils or fractions from pooled SP (each corresponding to 5% of total of fractionated material, or 0.75 ml SP) were diluted in 80 µl of PBS and then stained with ThT (final concentration 20 µM, from Sigma-Aldrich). Samples were incubated in the dark at room temperature for 15 min with constant shaking (350 rpm) before being measured in an Infinite M1000 Pro microplate reader (Tecan Group Ltd., Switzerland). Samples were excited at 435 nm and emission spectra were measured between 470 and 650 nm, with a 5 nm bandwidth and a manual gain set to 150.

## Microscopic analysis of endogenous amyloid interaction with spermatozoa

Endogenous amyloids from fractionated or unfractionated SP were visualized by staining samples with Proteostat and/or pFTAA, using approaches previously described (*Usmani et al., 2014*). Briefly, fractionated or unfractionated SP were incubated with Proteostat and/or pFTAA for 15 min at room temperature, and then transferred into an Ibidi Chamber Slide (#80826 from GmbH). Stained

samples were imaged on a Zeiss LSM710 AxioObserver confocal microscope equipped with a Plan-Apochromat 63/1.40 oil objective lens and Zen-Software v2010 (Zeiss, Germany). Spermatozoa were simultaneously imaged by DIC using transmitted light detector (T-PMT) and appropriate condenser settings. Where indicated, spermatozoa were additionally visualized by staining with Hoechst 33342.

## Single-cell sperm motility assay

Swim-up human spermatozoa were plated onto 5 mm coverslips (WPI, Sarasota, FL) in HEPES buffer solution (HS) containing 130 mM NaCl, 5 mM KCl, 1 mM $MgSO_4$, 2 mM $CaCl_2$, 5 mM glucose, 1 mM sodium pyruvate, 10 mM lactic acid, and 20 mM HEPES (pH 7.4). Coverslips with sperm cells were then placed in a recording chamber (Warner instruments, Hamden, CT) containing HS solution. Sperm movement was recorded at room temperature with a high-speed GX-1 Memrecam camera (NAC, Simi Valley, CA) attached to an Olympus IX71 microscope. Sperm motility was always confirmed before any peptide/fibril treatment and recorded. Following confirmation of sperm motility, HS containing 50 µg/ml of the appropriate peptide or amyloid fibril was perfused into the chamber, and then incubated with spermatozoa for 10 min. The speed of recording was 1000 frames per second (fps), but all supplemental videos were slowed down five-fold for clarity. Viability of immobilized spermatozoa was confirmed by monitoring mitochondrial activity with Mitotracker (Invitrogen, Grand Island, NY).

## Computer-assisted sperm analysis (CASA)

Motility at the population level was confirmed by CASA using methods previously described (*Mitra et al., 2010*). Peptides, fibrils, and SEM1 protein (G26-R281, from [*Silva et al., 2012*]) were all used at a final concentration of 50 µg/ml, while PBS alone served as the negative control. Sperm concentrations were diluted to 5,000/µl or 500/µl as indicated. The % motility inhibition mediated by SEM1(86–107) between the concentration range of 50–800 µg/ml was not statistically different, as high concentrations of the fibrils just caused larger aggregates to form, leading to immobilization of the same % of spermatozoa. Because each sperm sample had different % motility, each experiment was normalized by dividing each treatment by the % motility of the untreated sample and multiplying by 100. This set the % motility of the untreated sample to 100% and allowed comparison between different experiments using different sperm samples.

## Sperm viability assay

Human spermatozoa ($10^7$/ml) were treated with PBS or 50 µg/ml SEVI fibrils at 37°C for 30 min, and then incubated with 50 µg/ml of PI. Toxicity was assessed by monitoring uptake of PI by flow cytometry using a BD FACSCanto II (BD Biosciences, San Jose, CA).

## Sperm capacitation assays

Spermatozoa were assessed for the ability to capacitate by monitoring acrosome reaction and capacitation-mediated induction of tyrosine phosphorylation (*Arcelay et al., 2008*). Spermatozoa from fresh ejaculates were prepared by the swim-up method and then added to 6-well dishes containing 22 mm coverslips (WPI, Sarasota, FL) in HS buffer and allowed to settle for 30 min. Wells were then incubated with PBS, or 50 µg/ml SEM1(86–107) amyloid fibrils or the scrambled control, for an additional 15 min. Spermatozoa were washed with HS, and then capacitated for 4 hr at 37°C with HS in the presence of 25 mM $NaHCO_3$ and 20% fetal bovine serum (FBS). Cells were then either fixed with ice cold 95% ethanol for 30 min at 4°C for acrosomal staining, or lysed in 2X Laemmli sample buffer for Western blot analysis.

For acrosomal staining, spermatozoa fixed in 95% ethanol were allowed to air dry. Coverslips were then incubated with fluorescein isothiocyanate (FITC)-conjugated *Pisum sativum* agglutinin (Sigma-Aldrich) for 10 min, and then washed with ultra-pure water. Cells were then analyzed for acrosome reaction as described (*Lishko et al., 2010*).

For phosphotyrosine analysis, 4% $\beta$-mercaptoethanol ($\beta$-ME) was added to cells lysed in 2X Laemmli sample buffer. Lysates were boiled for 5 min at 100°C and then adjusted to a final $\beta$-ME concentration of 8%. Samples were transferred to 4–12% polyacrylamide gels and blotted onto PVDF membranes. Membranes were blocked with 3% γ-globulin-free BSA in PBS containing 0.1% Tween (PBS-T) for 30 min at room temperature. Subsequently, membranes were incubated for 1 hr

at room temperature with mAB 4G10 Platinum, anti-phosphotyrosine (Millipore, Billerica, MA) or anti-acetylated alpha-tubulin (Thermo Scientific, Waltham, MA) diluted 1:5000 in PBS-T containing 1% γ-globulin-free BSA. After washing three times with PBS-T, the membranes were incubated for 1 hr at room temperature with goat anti-mouse HRP-conjugated IgG (Millipore) diluted 1:20,000 in PBS-T containing 1% γ-globulin-free BSA. Protein bands were detected by SuperSignal West Pico Chemiluminescent Substrate (Thermo Scientific) on a FluorChem M imaging system (Protein Simple, Santa Clara, CA).

## Assessment of phagocytic activity of macrophages

Phagocytic activity of macrophages was assessed using the Vybrant Phagocytosis Assay Kit (Thermo Fisher). Peripheral blood mononuclear cells (PBMCs) were isolated from fresh TrimaLeuko reduction chambers from female donors (obtained through the Blood Centers of the Pacific Blood Systems) using Ficoll-Hypaque density gradients. CD14+ monocytes were then purified using CD14+ microbeads (Miltenyi Biotec, Bergisch Gladbach, Germany). Monocytes were differentiated into macrophages by culturing $10^6$ cells/well in clear flat-bottom 6-well plates (Corning Primaria, Corning, NY) in RPMI supplemented with 10% FBS, 2 mM L-glutamine, 50 U/ml penicillin, and 50 µg/ml streptomycin, in the absence or presence of 100 ng/ml human recombinant MCSF (R&D Systems, Minneapolis, MN). Five days later, cells were fed with fresh media, and 48 hr later all cultures were replaced with fresh media lacking MCSF. The cells cultured in the absence or presence of MCSF were assessed for phagocytic activity on dead fluorescent bacteria following guidelines provided by the manufacturer.

## Sperm labeling

For analysis by ImageStream, swim-up cells were centrifuged at 526 x g and then the pellet containing the sperm cells was resuspended in 0.05 µM Cell Proliferation Dye eFluor 670 (eBioscience, San Diego, CA). For flow cytometry experiments, a concentration of 2.5 µM eFluor 670 was instead used. Sperm cells were labeled for 10 min at room temperature and washed three times with PBS prior to use in phagocytosis assays. To compare healthy (fresh and motile) vs. damaged spermatozoa, the sperm cells were evenly divided following fluorescence labeling with eFluor 670. One half, containing the healthy and motile spermatozoa, was incubated at room temperature while the other half was damaged by five rounds of freezing/thawing using liquid nitrogen. A complete loss of motility in the freeze/thawed but not the mock-treated sample was confirmed by visual inspection by light microscopy. Alternatively, to induce apoptosis, swim-up spermatozoa were incubated for 24 hr at 25°C after eFluor 670 labeling.

To simultaneously monitor two populations of spermatozoa within the same sample, two different dyes were used. Unstained sperm cells were isolated using approaches described above and divided into two equal parts. The first population was labeled with eFluor 670 as described above, while the second population was labeled with a 160 µM Celltracker Blue CMAC (Thermo Fisher) for 1 hr at 37°C. The CMAC labeled cells were then washed twice with PBS and were then damaged by five rounds of freeze/thaw using liquid nitrogen.

## Sperm phagocytosis assay

Macrophages were differentiated from monocytes ($10^6$ cells / condition) in the presence of MCSF as described above. A total of 1.5–2 $\times$ $10^6$ spermatozoa for each treatment condition were incubated in the absence or presence of 100 µg/ml SEM1(86–107) fibrils for 15 min at 37°C, and then added to the macrophages. Unlabeled spermatozoa were used for Western blot experiments, while spermatozoa labeled using methods described above were used for flow cytometry and ImageStream experiments. Phagocytosis was allowed to proceed for the designated amount of time (0.5, 1.5, 3, or 6 hr) at 37°C or 4°C as appropriate. At each timepoint, macrophages were washed three times with RPMI and treated with 2.5 µg/ml of Heparin (Sigma-Aldrich) in RPMI for 0.5 hr at 37°C to help remove surface-bound fibrils. Following three additional washes with PBS, cells were either lysed for Western blot analysis, or stained for phenotyping by flow cytometry or Imagestream. For comparing phagocytosis of spermatozoa with different proportions of apoptotic spermatozoa, eFluor 670 stained spermatozoa were either added to the macrophages immediately after being stained and washed, or they were incubated at room temperature for 24 hr to induce apoptosis prior to being added to

the macrophages. Phagocytosis of these spermatozoa was allowed to proceed for 0.5 hr and then analyzed by flow cytometry.

## Western blot to assess phagocytosis

At the designated timepoint, macrophages were lysed in 25–35 µl 2X Laemmli's buffer (263 mM Tris pH 6.8, 80% glycerol, 142 mM of SDS, 29 mM of bromophenol blue), containing protease inhibitor cocktail (Roche, Basel, Switzerland) and 0.1 mM PMSF. Lysates were heated for 5 min at 97°C immediately before loading 20 µl on a 7.5% Criterion Tris-HCl polyacrylamide gel (Bio-Rad, Hercules, CA). Proteins were transferred for 2 hr at 65 V onto nitrocellulose Immuno-Blot PVDF membranes (Bio-Rad). Membranes were then blocked for 0.5 hr with 3% BSA in PBS-Tween (PBS-T), incubated at a 1:2000 dilution with a primary antibody against acetylated tubulin (Sigma-Aldrich) which exhibits specificity for sperm cells (*Piperno and Fuller, 1985*), followed by washing and a secondary anti-mouse sheep IgG HRP antibody at a 1:5000 dilution (GE Healthcare, Chicago, IL). Immunoblots were developed using Western Lightning ECL (Perkin Elmer, Waltham, MA).

## Flow cytometry and imagestream assessment of sperm phagocytosis

At the appropriate timepoint, macrophages were trypsinized using 1 ml of 0.05% Trypsin containing 0.53 mM EDTA, washed, and then incubated for 0.5 hr at 4°C with anti-CD14 (Clone M5E2, conjugated to FITC) and anti-CD33, (Clone WM53, conjugated to PE). The samples were then washed, fixed with 1% PFA, and run on an LSRII flow cytometer (Becton Dickinson, Franklin Lakes, NJ). Macrophages were identified on the LSRII by gating on CD14+CD33+ cells. For Imagestream analysis, cells were identified by CD14 expression. Negative controls for fluorescence labeling and autofluorescence included the following: unlabeled sperm cells, labeled sperm cells, SEM1(86–107) fibrils, labeled sperm cells + SEM1(86–107) fibrils, macrophages, and macrophages + SEM1(86–107) fibrils.

## Confocal microscopy analysis of sperm phagocytosis

Monocyte-derived macrophages were first stained with 2.5 µg/ml of Vybrant DiO Cell-Labeling Solution (Thermo Scientific) diluted in PBS. The cells were then washed three times with PBS and incubated with eFluor 670 spermatozoa in the absence or presence of SEM fibrils using conditions described above. After 3 hr, macrophages were washed 3x with PBS, trypsinized, and stored in 1% PFA at 4°C in the dark until imaging. Confocal imaging was carried out using a Nikon Eclipse Ti-E inverted microscope equipped with a Yokogawa CSU22 spinning disk confocal scanner, a Sutter emission Lambda filter wheel adapter with ET460/50m, ET525/50m, ET645/65m and ET700/75m filters, a Prior motorized stage with Piezo Z-drive, and a Photometrics Evolve 512 Delta EMCCD Camera. Images were acquired with Micro-Manager software version 1.4.21. Image scanning was executed employing a Plan Apo VC 100x/1.4 Oil (DIC N2/100X I) objective using 488 nm and 640 nm, 100 mW Coherent OBIS lasers. The Piezo Z-drive was set for fast acquisition and z-steps at 0.40 µm. Image analysis was performed using ImageJ software version 1.48 with the Deconvolution lab (EPFL) and 3D viewer (B. Schmid) plugins installed. All acquisition and analysis parameters were maintained constant for all samples.

## Quantitating apoptotic sperm cells

Fresh spermatozoa and spermatozoa incubated for 24 hr at 25°C were stained for the apoptotic marker phosphatidylserine by use of FITC-conjugated Annexin V (eBioscience) followed by flow cytometric analysis on an LSRII (Becton Dickinson), following manufacturer's protocol.

## Acknowledgements

We thank T Wirth at the Transgenic Mice Core Facility for helping with IVF assays; J Wong at the Gladstone Electron Microscopy Core Facility for generating the electron micrographs; N Mannowetz for assistance in sperm motility recording; AL Lucido for editorial assistance; T Roberts and C Goodfellow for assistance in preparing the figures; and S Cammack and R Givens for administrative assistance. This work was supported by the National Institutes of Health (K99AI104262, R00AI104262, R21AI116252, and R01AI127219 to NRR; pilot grant to NRR as part of parent grant P50HD055764 to Linda C Giudice; R01HD074511 to CP; and P01 AI083050 to WCG), the Deutsche

Forschungsgemeinschaft (DFG) (MU 3115–2 to JM), and the European Research Council (ERC-AG-LS6 to FK). As part of a parent grant to Paul A Volberding, the National Institutes of Health also funded the grant P30AI027763 which provided money for instruments used in the paper. The Amnis Imagestream used in this study was funded by the Department of Defense (W81XWH-11-1-0562).

## Additional information

### Funding

| Funder | Grant reference number | Author |
| --- | --- | --- |
| National Institutes of Health | K99AI104262 | Nadia R Roan |
| National Institutes of Health | R00AI104262 | Nadia R Roan |
| National Institutes of Health | R21AI116252 | Nadia R Roan |
| National Institutes of Health | R01AI127219 | Nadia R Roan |
| National Institutes of Health | R01HD074511 | Christopher D Pilcher |
| National Institutes of Health | P01 AI083050 | Warner C Greene |
| U.S. Department of Defense | W81XWH-11-1-0562 | Warner C Greene |
| Deutsche Forschungsgemeinschaft | MU 3115-2 | Jan Münch |
| European Research Council | ERC-AG-LS6 | Frank Kirchhoff |

The funders had no role in study design, data collection and interpretation, or the decision to submit the work for publication.

### Author contributions

NRR, Conceptualization, Formal analysis, Supervision, Funding acquisition, Investigation, Methodology, Writing—original draft, Project administration, Writing—review and editing; NS-M, Conceptualization, Investigation, Methodology; NK, KGH, JR, Formal analysis, Investigation; SMU, Conceptualization, Investigation; JN, MM, OS, Investigation, Methodology; LS, AR, Formal analysis, Investigation, Methodology; MC, Conceptualization, Formal analysis, Funding acquisition, Writing—review and editing; KM, JFS, FG, Resources; CDP, Resources, Funding acquisition; MO'R, Conceptualization, Formal analysis, Investigation, Methodology, Writing—review and editing; PVL, Conceptualization, Resources, Formal analysis, Methodology, Writing—review and editing; FK, Conceptualization, Writing—review and editing; JM, Conceptualization, Formal analysis, Supervision, Funding acquisition, Writing—review and editing; WCG, Conceptualization, Funding acquisition, Writing—review and editing

### Author ORCIDs

Nadia R Roan, http://orcid.org/0000-0002-5464-1976
Katherine G Hamil, http://orcid.org/0000-0001-7430-8313
Polina V Lishko, http://orcid.org/0000-0003-3140-2769

### Ethics

Human subjects: De-identified fresh human semen samples were obtained from University of California, San Francisco (UCSF) / San Francisco General Hospital (SFGH) Positive Health Program Research Group (The Options Project, IRB # 10-00301) and the Kinderwunsch-Zentrum (Ulm, Germany) after giving informed consent.

Animal experimentation: The use of animals was approved by the Regierungspräsidium Tübingen, registration number 0.185, and was in accordance with existing regulations of the German Federal Law on Care and Use of Laboratory Animals.

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
