## [Decision Letter]

Thank you for submitting your article "Semen amyloids participate in spermatozoa selection and clearance" for consideration by *eLife*. Your article has been reviewed by two peer reviewers, and the evaluation has been overseen by Randy Schekman as the Senior Editor. The reviewers have opted to remain anonymous.

The reviewers have discussed the reviews with one another and the Reviewing Editor has drafted this decision to help you prepare a revised submission.

Summary:

Roan et al. explore the potential function of semen amyloids in clearance of mammalian sperm from the female reproductive tracts. There is a strong biological rationale that a clearance system of some sort must exist. We have known for several decades that this system includes a female leukocyte response, but the mechanisms that account for clearance are poorly understood. The identification of a role of amyloid fibrils from seminal peptides in sperm trapping and clearance is a novel advance. These data also add to an emerging literature on a functional (that is, non-pathological) role of amyloid fibrils (eg, PMID: 19622831) and so may be of wider interest.

In general, the experiments are well done and carefully analyzed. These, and the proposed mechanism of seminal amyloid, are sufficiently novel and interesting to merit publication in *eLife*.

Essential revisions:

This manuscript proposes the interesting idea that amyloid fibrils in semen participate in sperm selection and clearance. The studies follow previous work by these investigators in which amyloid fibrils in semen were suggested to promote HIV infection. Functional amyloids have been identified in a broad range of cells and tissues including the male reproductive tract and several studies indicate a role for amyloids in innate immunity and thus a role in sperm quality control is certainly a possibility. However, the data do not provide clear evidence that amyloid fibrils in semen have these functions.

The primary concern I have with these studies is that all experiments were performed with amyloid fibrils that were generated in vitro from chemically synthesized peptides and thus may not be representative of what is present in vivo. Because of this, the use of "semen fibrils" throughout the manuscript is misleading since unless the methods are carefully read, the reader gets the incorrect impression that biological samples were used when in fact they were not. It is unlikely that such "pure" amyloids (never exposed to other proteins, lipids etc) exist in semen and indeed the thought that semen amyloids likely represent fibrils formed from multiple peptides was previously suggested by the authors. The studies would have been strengthened considerably by examining amyloids isolated from semen as described by the authors in previous studies (Nature Communications 2014). The addition of conformation-dependent antibodies (A11/OC) during IVF or sperm incubation experiments to determine if outcomes were blocked(sperm immobilization or phagocytosis) would lend further support that amyloids mediate these processes and might also shed light on the form of amyloid that is involved. A second concern is the authors' assumption they are looking solely at the effects of amyloid fibrils rather than also the oligomeric forms (or only the oligomeric forms) of the SEV1/SEM peptide amyloids. In their methods they state that amyloid formation was confirmed by EM and ThT analyses. However, these data are not shown; moreover these approaches do not allow for a quantitative assessment if, in addition to fibrils, other forms of amyloid such as oligomers are present. This could be done by dot blot using conformation-dependent antibodies A11/OC which recognize oligomeric and fibrillar forms, respectively. Similarly, the authors conclusion that because Abeta amyloid did not immobilize sperm suggested that the semen fibrils were distinct in their ability for this function could also be due to there being less oligomeric amyloid forms in the Abeta preparation compared to the semen fibrils.

[Editors' note: further revisions were requested prior to acceptance, as described below.]

Thank you for resubmitting your work entitled "Semen amyloids participate in human spermatozoa selection and clearance" for further consideration at *eLife*. Your revised article has been favorably evaluated by Randy Schekman, Senior editor and Reviewing editor and one reviewer.

The manuscript has been improved but there are some remaining issues that need to be addressed before acceptance.

This studies presented in this manuscript suggest that amyloids in human semen participate in sperm selection and clearance. The addition of new data showing endogenous amyloids, like those generated from synthetic peptides, trap and immobilize sperm, and the use of conformation-dependent antibodies to establish that several amyloid forms are present in their preparations strengthen the manuscript from its previous submission and provide evidence for a new functional amyloid.

Except for one minor clarification that is needed, the manuscript is acceptable for publication:

For this reader it is still not clear what the distinction is between the SEV1 and SEM peptide fibrils as they are not defined in the Introduction. The text states "two classes of semen amyloids have been identified: those derived from PSA-generated fragments of prostatic acid phosphatase (PAP) and those derived from PSA-generated fragements of semenogelins". Are the SEV1 derived from the PAP and the SEM derived from semenogelins?

---

## [Author Response]

*Essential revisions:*

*This manuscript proposes the interesting idea that amyloid fibrils in semen participate in sperm selection and clearance. The studies follow previous work by these investigators in which amyloid fibrils in semen were suggested to promote HIV infection. Functional amyloids have been identified in a broad range of cells and tissues including the male reproductive tract and several studies indicate a role for amyloids in innate immunity and thus a role in sperm quality control is certainly a possibility. However, the data do not provide clear evidence that amyloid fibrils in semen have these functions.*

*The primary concern I have with these studies is that all experiments were performed with amyloid fibrils that were generated* in vitro *from chemically synthesized peptides and thus may not be representative of what is present* in vivo*. Because of this, the use of "semen fibrils" throughout the manuscript is misleading since unless the methods are carefully read, the reader gets the incorrect impression that biological samples were used when in fact they were not. It is unlikely that such "pure" amyloids (never exposed to other proteins, lipids etc) exist in semen and indeed the thought that semen amyloids likely represent fibrils formed from multiple peptides was previously suggested by the authors.*

The reviewer brings up a good point that synthetic amyloid fibrils may have different properties compared to the endogenous form of the amyloid. With regards to semen fibrils, however, we’d like to clarify that endogenous SEVI and SEM fibrils are present in semen, and that synthetic and endogenous semen fibrils are similar in structure, charge, and HIV-enhancing activity (Usmani et al., 2014; Roan et al., 2014). These observations suggest that endogenous amyloids are unlikely to be markedly complexed with other proteins and lipids, or at least do not need to be so to exert physical properties similar to that of the synthetic versions of these fibrils.

*The studies would have been strengthened considerably by examining amyloids isolated from semen as described by the authors in previous studies (Nature Communications 2014). The addition of conformation-dependent antibodies (A11/OC) during IVF or sperm incubation experiments to determine if outcomes were blocked(sperm immobilization or phagocytosis) would lend further support that amyloids mediate these processes and might also shed light on the form of amyloid that is involved.*

Notably, in our previous characterization of endogenous semen amyloids (Usmani et al., 2014), we did not isolate endogenous amyloids from semen, but characterized endogenous amyloids in unfractionated liquefied semen using microscopic approaches. Since the purification of endogenous amyloids from semen is very technically challenging and requires large volumes of semen, in this study we had performed experiments with synthetic fibrils to have sufficient amounts of material for a comprehensive and detailed examination. However, we agree with the reviewer’s comment that the novel function of semen fibrils described in our manuscript should be confirmed with endogenous material. Therefore, we collected 26 ml of semen pooled from 20 donors, and by fractionation purified endogenous amyloids as confirmed by staining with various amyloid-binding dyes (thioflavin T, Proteostat, pFTAA, please see the new Figure 2—figure supplement 2 and Figure 2—figure supplement 3). Despite the limiting quantities of these endogenous amyloids and use of much of the material for the microscopic studies to prove successful isolation of amyloids, we were able to perform a small set of experiments examining the interplay of these amyloids with sperm.

First, using microscopy, we show that purified endogenous amyloids bind to sperm (new Figure 2—figure supplement 3). This result confirms and extends our finding that endogenous amyloids also interacted with sperm in unfractionated semen (new Figure 2—figure supplement 3). Second, using the sperm entrapment assay, we show that purified endogenous amyloids, like their synthetic counterparts, trap and immobilize sperm (new panel Figure 2). Thus, endogenous and synthetic amyloids display similar activity in these various assays. We have made clear throughout the manuscript which experiments were performed with synthetic fibrils and which involved endogenous amyloids.

We believe these new data with endogenous semen fibrils strengthens our paper. Also, when these results are taken together with previously published data that endogenous semen fibrils share the same biophysical properties as synthetic semen fibrils (Usmani et al., 2014; Roan et al., 2014), we believe they sharply counter the argument that the observed effects are an artifact of in vitro fibril production. Unfortunately, we could not test the effects of A11/OC in this system due to a lack of sufficient endogenous material. Whether both fibrils and oligomers exert the reported biological effects remains an interesting question that we propose to evaluate in the future.

*A second concern is the authors' assumption they are looking solely at the effects of amyloid fibrils rather than also the oligomeric forms (or only the oligomeric forms) of the SEV1/SEM peptide amyloids. In their methods they state that amyloid formation was confirmed by EM and ThT analyses. However, these data are not shown;*

*Moreover these approaches do not allow for a quantitative assessment if, in addition to fibrils, other forms of amyloid such as oligomers are present. This could be done by dot blot using conformation-dependent antibodies A11/OC which recognize oligomeric and fibrillar forms, respectively.*

We have conducted the requested experiment (new Figure 1—figure supplement 2), which demonstrated that A11 and OC bind to structures in seminal plasma, synthetic SEVI fibrils, and synthetic SEM fibrils, but not the corresponding monomeric peptides nor serum plasma (used as a negative control). These data suggest that seminal plasma and synthetic fibrils harbor fibrils and fibrillar oligomers (recognized by OC) as well as prefibrillar oligomers (recognized by A11).

*Similarly, the authors conclusion that because Abeta amyloid did not immobilize sperm suggested that the semen fibrils were distinct in their ability for this function could also be due to there being less oligomeric amyloid forms in the Abeta preparation compared to the semen fibrils.*

We thank the reviewer for pointing this out. Indeed, it is possible that Abeta did not efficiently immobilize due to different amounts of oligomers in these preparations. We have now added a statement in the Results section acknowledging this possibility.

[Editors' note: further revisions were requested prior to acceptance, as described below.]

*[…] Except for one minor clarification that is needed, the manuscript is acceptable for publication:*

*For this reader it is still not clear what the distinction is between the SEV1 and SEM peptide fibrils as they are not defined in the Introduction. The text states "two classes of semen amyloids have been identified: those derived from PSA-generated fragments of prostatic acid phosphatase (PAP) and those derived from PSA-generated fragements of semenogelins". Are the SEV1 derived from the PAP and the SEM derived from semenogelins?*

We are very pleased that upon resubmission the manuscript was deemed acceptable for publication pending one point of clarification regarding the distinction between SEVI and SEM fibrils. The editor/reviewer asked whether SEVI fibrils are derived from PAP and SEM fibrils from the semenogelins. This is indeed the case and to clarify this point, we have added statements in the sentence in the Introduction that describes these amyloids.